# Effect of Diurnal Variation of Heart Rate and Respiratory Rate on Activation of Rapid Response System and Clinical Outcome in Hospitalized Children

**DOI:** 10.3390/children10010167

**Published:** 2023-01-14

**Authors:** Lia Kim, Kyoung Sung Yun, June Dong Park, Bongjin Lee

**Affiliations:** Department of Pediatrics, Seoul National University College of Medicine, Seoul 03080, Republic of Korea

**Keywords:** vital signs, heart rate, respiratory rate, early warning score, child

## Abstract

Heart rate and respiratory rate display circadian variation. Pediatric single-parameter rapid response system is activated when heart rate or respiratory rate deviate from age-specific criteria, though activation criteria do not differentiate between daytime and nighttime, and unnecessary activation has been reported due to nighttime bradycardia. We evaluated the relationship between rapid response system activation and the patient’s clinical outcome by separately applying the criteria to daytime and nighttime in patients < 18. The observation period was divided into daytime and nighttime (8:00–20:00, and 20:00 to 8:00), according to which measured heart rate and respiratory rate were divided and rapid response system activation criteria were applied. We classified lower nighttime than daytime values into the ‘decreased group’, and the higher ones into the ‘increased group’, to analyze their effect on cardiopulmonary resuscitation occurrence or intensive care unit transfer. Nighttime heart rate and respiratory rate were lower than the daytime ones in both groups (both *p* values < 0.001), with no significant association with cardiopulmonary resuscitation occurrence or intensive care unit transfer in either group. Heart rate and respiratory rate tend to be lower at nighttime; however, their effect on the patient’s clinical outcome is not significant.

## 1. Introduction

Heart rate (HR) and respiratory rate (RR) are key indicators of the patient’s physiological status: patients generally present with changes in vital signs before physiological deterioration. Therefore, HR and RR are used as evaluation factors in most pediatric screening systems, such as pediatric advanced life support (PALS), advanced pediatric life support (APLS), rapid response system (RRS), and pediatric early warning score (PEWS) to identify patients who need intervention [1,2,3,4,5,6,7].

On the other hand, it is well known that HR and RR may have circadian variations under the influence of the autonomic nervous system. Indeed, some studies report lower HR and RR at nighttime [6,7,8,9,10,11]. However, the criterion of HR or RR by age used in the above-mentioned screening systems does not differentiate between daytime and nighttime [3,4,5,6,7]. However, due to their characteristics, RRS and PEWS are applied not only during the daytime but should also be applied at nighttime. Thus, false positives may be recorded since the HR or RR is normally lowered at nighttime, which can be interpreted as a patient’s worsening. While PEWS scores and evaluates various parameters, RRS relies on a single parameter, providing early warning only based on an outlier of one factor (HR or RR), whereby the probability of false positives may be higher. In one retrospective study, in the case where RRS was activated only with bradycardia, no significant association was found with the deterioration of the patient. This study also reported more cases of RRS activation due to nighttime bradycardia than during the daytime, suggesting that attention should be paid to the application and evaluation of vital sign monitoring according to the time period [12].

However, studies so far have only focused diurnal variation in HR and RR or the underlying mechanisms [8,9,10,11], none of them investigating the relationship between the differences in vital signs between daytime and nighttime and the related clinical outcomes. Therefore, this study aimed to obtain the distribution of HR and RR for nighttime and daytime to investigate the clinical significance of the changes in HR or RR meeting the RRS activation criteria.

## 2. Materials and Methods

### 2.1. Study Setting and Design

This retrospective study was conducted at a tertiary care children’s hospital with 350 beds. Children under the age of 18 who were admitted to the general ward at the children’s hospital from January 2019 to December 2020 were included the study. Among them, patients diagnosed with cardiovascular disease or pulmonary disease that could affect HR or RR were excluded from the analysis. Since body temperature is also known to affect HR and RR, cases with a body temperature of <36 °C or ≥38 °C at the time of vital sign measurement were excluded [13]. Among the HR and RR measurements, those obtained in the intensive care unit (ICU), operating room, or emergency department, but not in the general ward, were excluded from the analysis.

The source of the data is the clinical data warehouse of the hospital information system, which was accessed following the deliberation of the hospital’s institutional review board (IRB). In addition, as this study analyzed de-identified data, the requirement for written consent was waived from the IRB (2112-151-1285). We collected data on patient gender, HR, RR, body temperature, measurement time, activation of RRS, cardiopulmonary resuscitation (CPR) occurrence, and transfer to ICU.

As a pre-processing of the collected data, observations considered to be non-physiologic data (with HR > 300 beats/min, HR < 30 beats/min, RR > 120 breaths/min, or RR < 5 breaths/min) were excluded. The daytime period for measuring vital signs was defined from 8:00 to 20:00, and the nighttime period as that from 20:00 to 8:00 (of the next day). A patient is repeatedly measured multiple times due to the characteristics of vital signs; however, all individual measurements were not used in the analysis, to avoid bias in the results attributable to differences in the number of times the vital signs were measured in some patients compared with others. Each patient’s hospitalization period was defined as an ‘individual hospitalization unit’, and the mean of HR and RR for each unit was used for analysis. The confounding effect of the variability of subjects with respect to their daytime vs. nighttime difference in HR and RR was eliminated using only the daytime–nighttime pair of measurements of the same patient’s individual hospitalization unit in the analysis. Thus, we excluded cases with HR or RR measured during the daytime, but not measured at nighttime, or vice versa.

### 2.2. RRS Activation Criteria and Classification of Abnormalities in Measurements

Since October 2010, this institution has employed the RRS system by slightly modifying the system introduced by Tibballs et al. [12,14,15]. In this study, the original RRS criteria were used to evaluate whether they corresponded to the activation criteria of HR and RR, and each measured value was classified into one of the three categories: low, normal, and high. In addition, we identified a ‘decreased group’ and an ‘increased group’ according to whether the values measured for nighttime were lower or increased compared to those for daytime: the values classified as low in the nighttime but not low in the daytime (normal or high) were included in the decreased group, and those classified as high in the nighttime but not high (normal or low) in the daytime were included in the increased group. HR difference was obtained by subtracting daytime HR from nighttime HR, and RR difference was obtained in the same way. In addition, we defined CPR occurrence or transfer to ICU as a critical event to represent the patient’s clinical outcome, and attempted to analyze its relevance according to the classification group of measurements.

### 2.3. Outcomes

The primary outcome was to evaluate the association of critical events (CPR occurrence or transfer to ICU) according to the HR and RR groups (increased group or decreased group). The secondary outcomes were HR and RR differences, and distribution and centiles of HR and RR by daytime and nighttime. In addition, since HR and RR have different normal ranges depending on age, the meaning of z-scores by age may be different than that of the respective individual measurements. Thus, the distribution and difference of z-scores were also analyzed.

### 2.4. Statistical Analyses

Categorical variables were expressed as numbers and percentages, continuous variables were expressed as mean (standard deviation [SD]) if they followed normal distribution, and non-parametric variables were denoted by median (interquartile range [IQR]). The association with the critical events of each group was analyzed using the mixed effect logistic regression model. The results were expressed as odds ratio (OR) and 95% confidence interval (CI). Paired t-test or Wilcoxon signed-rank test was used for comparison of daytime and nighttime measurements depending on normality. The Shapiro–Wilk test was used for the normality test. For the calculation of z-scores by age for HR and RR, the Lambda-Mu-Sigma method and the Box–Cox power exponential distribution were used based on the data derived from our previous study that presented centiles of HR and RR as a nationwide study [16]. In this process, the generalized additive model for location, scale, and shape package, and super imposition by translation and rotation growth curve analysis package were used [17,18]. R software version 4.2.1 was used for all data processing and statistical analyses. *p* values < 0.05 were considered statistically significant.

## 3. Results

### 3.1. Clinical Characteristics of the Patients

There were 11,890 hospitalizations in a total of 3824 patients, and 190,133 vital signs were measured. The 9778 individual hospitalization units in a total of 3633 patients were finally used for analysis after applying the exclusion criteria. The age was 7.1 (2.7–12.2) (median [IQR]) years old, and 4394 (44.9%) were female patients. Among the individual hospitalization units, the RRS was activated in 1021 (10.4%) cases, 541 (5.5%) cases were transferred to the ICU, and CPR occurred in 69 (0.7%) cases. Other demographic data are detailed in Table 1.

As for the underlying disease of patients, hemato-oncologic disease was the most common with 5665 (21%) patients, followed by congenital and genetic disease and neurologic disease, with 4670 (17.3%) and 2126 (7.9%) patients, respectively (Appendix A). Distributions of HR and RR according to time period are shown in Appendix A.

### 3.2. Main Outcomes

CPR occurrence and ICU transfer were analyzed via group-specific critical event analysis, which is the primary outcome of this study. RRS activation was also analyzed. In the case of CPR incidence, the results in all four groups (decreased HR group, decreased RR group, increased HR group, and increased RR group) were not statistically significant (ORs [95% CI] 0.595 [0–49,800,930.118], 0 [not applicable], 0 [not applicable], and 61.58 [0.013–296,863.903], respectively). ICU transfer also showed statistically insignificant results in all groups (ORs [95% CI] 0.495 [0.177–1.389], 0 [not applicable], 0 [not applicable], and 0.078 [0.001–10.379], respectively). On the other hand, RRS activation showed statistically significant results in the decreased HR group (OR [95% CI] = 4.928 [3.406–7.129]), increased HR group (OR [95% CI] = 4.429 [1.155–16.990]), and increased RR group (OR [95% CI] = 7.608 [1.788–32.367]), but not in the decreased RR group (OR [95% CI] = 2.366 × 1037 [not applicable]) (Table 2).

The HR difference and RR difference are shown in Figure 1.

The values in the young age part of these scatter plots were widely distributed, showing a trend of narrowing as the age increased. As a result of regression analysis, both the SD of HR difference (*p* < 0.001) and SD of RR difference (*p* < 0.001) showed a statistically significant decrease with increasing age (Appendix A). Additionally, the centile curves and charts of HR and RR by time period are presented in supplementary information (daytime HR centile curve and chart: Appendix A; nighttime HR: Appendix A; daytime RR: Appendix A; and nighttime RR: Appendix A).

As for the z-score by age according to the time period, daytime showed higher results than nighttime in both HR and RR (both *p* values < 0.001) (Figure 2). HR and RR differences according to underlying disease are shown in Appendix A.

When the RRS activation criteria were applied to HR and RR by time period, daytime HR was 0.8%, 98.8%, and 0.4% for low, normal, and high, respectively, and for nighttime, 3.2%, 96.6%, and 0.2%, respectively. A high ratio of low HR was shown (Figure 3A,B).

On the other hand, RR was 0.0%, 99.5%, and 0.4% at daytime versus 0.0%, 99.6%, and 0.4% at nighttime, showing similar results (Figure 3C,D). In the same way, the distributions to which APLS and PALS criteria are applied instead of RRS activation criteria are shown in Appendix A, respectively.

**Figure 3 children-10-00167-f003:**
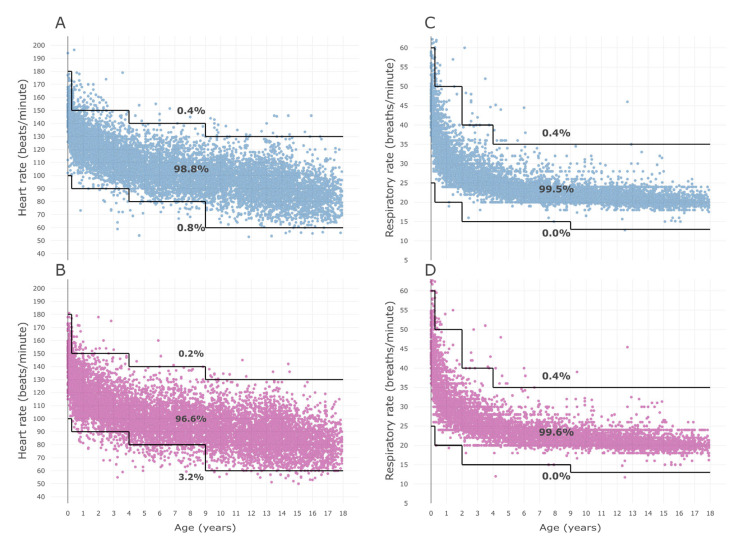
Distribution of HR and RR by time period and RRS activation criteria by age. (**A**) HR distribution during daytime, (**B**) nighttime HR, (**C**) daytime RR, and (**D**) nighttime RR distribution. Blue dots indicate daytime (from 8:00 to 20:00) measurements, and red dots indicate nighttime (from 20:00 to 8:00 the next day) measurements. The solid black line represents the age-specific criteria for RRS activation [12], and the percentages represent the percentages of the measured values in each range. HR = heart rate, RR = respiratory rate, RRS = rapid response system.

## 4. Discussion

We conducted this study to evaluate the clinical significance of RRS activation due to diurnal variation in HR or RR and revealed that neither the decreased group nor increased group had a significant effect on CPR occurrence or ICU transfer. This result allowed us to consider several interesting points.

There was a statistically significant difference in HR and RR during the daytime and nighttime, especially a decline during the nighttime compared to daytime. This is consistent with the results of several previous studies on the diurnal variation in vital signs [8,11,14,20,21]. One study suggested that the circadian rhythm of HR was related to autonomic nervous activity, such as the relative degree of sympathetic tone or parasympathetic tone [8]. Another study on vital sign abnormalities in children showed similar results to the above study, suggesting that vagal slowing and sleep can be the cause of bradycardia or bradypnea [14].

However, one interesting fact is that, due to the diurnal variation of vital signs, if the RRS activation criteria are met only at nighttime, the possibility that this identifies a condition becoming life-threatening is low. Of course, this does not mean that HR or RR outside the normal range are not clinically significant. However, if it only deviates from the normal range at night due to physiologic diurnal variation, it gives only a small hint as to how to a primary physician facing this case should respond. This also has implications for the application of RRS activation criteria during nighttime. Haines et al. indicated that bradycardia alone was not a specific marker for serious illness due to its low specificity [15]. Another study on pediatric RRS indicated the limitations of using bradycardia as a single parameter, and suggested the need for other co-parameters to activate RRS [12]. On the other hand, in contrast to these results for bradycardia, another previous study reported that 25% of patients admitted to ICU had tachypnea [16]. However, care should be exercised in interpreting this study, because it literally reports the rate of tachypnea among patients admitted to the ICU, and not an analysis of the relationship between ICU admission and tachypnea, since the latter was seen only at night while RR was normal during the day.

On the other hand, the SD of the difference in HR or RR was higher at a younger age and decreased with increasing age (Appendix A). This means that younger pediatric patients show an inherently greater variation in HR and RR, thus suggesting that there may be limitations in interpreting and evaluating the patient’s condition with only one change in HR or RR in younger children, such as infants. Although it is true that the median of HR or RR is higher at a young age [19,21,22], this may not mean that the lower limit of HR or RR should be higher with younger age. This is because the difference in HR or RR increases significantly as the age decreases. Therefore, caution is required when interpreting vital signs, especially in young children.

Single-parameter RRS (used in this study) triggered by meeting the criteria for only one parameter may be more vulnerable to diurnal variation of vital signs compared to PEWS, which is scored by a combination of several parameters [12]. However, even in PEWS, since diurnal variation is not considered in the criteria of HR or RR, the same intrinsic limitations exist as in RRS, although their effect is smaller. Nevertheless, we do not think that it would be best to present differentiated reference ranges according to daytime and nighttime to apply RRS or PEWS. The accuracy would be slightly increased; however, we need to consider the improvement in the quality of medical care in terms of the overall cost and benefit of medical resources. Nevertheless, we believe that these individual studies, such as ours, can be gathered to create medical evidence and ultimately be used as a foundation for medical development. There have been several studies that tried to derive centile curves of HR and RR based on actual evidence [19,21,22,23,24], revealing how different actual evidence is from the age criteria used in PALS and APLS [19,21]; however, the criteria for HR and RR of PALS were not changed following such evidence. All the same, these studies are not meaningless at all, as they broaden the medical knowledge base and solidify the thinking of researchers; we do hope that our research work will contribute along this line.

A key strength of our study is that we analyzed the distribution of HR and RR according to the time period. Previous studies mentioned diurnal variations in vital signs, though not reflecting their effect on reference ranges. To the best of our knowledge, this is the first study to analyze each distribution by daytime and nighttime period, and to evaluate the clinical significance of changes in vital signs in children.

This study has several limitations. First, this was a single-center study reflecting the characteristics of the patients admitted to our hospital. It may be difficult to generalize our results to other centers. Second, we analyzed results by the average value of HR and RR for individual hospitalization unit. Since the analysis was conducted using representative values (average) rather than the individual values measured, there might have been a loss in the signal regarding the changes in individual vital signs. However, patients with a long hospital stay and a lot of vital sign measurements could have skewed the results if the analysis had been done using individual measured values, thus it was inevitable to use representative values in order to minimize such bias. Third, a more accurate analysis could have been possible if the severity of patients transferred to the ICU had been corrected through severity score, such as pediatric risk of mortality score or pediatric index of mortality-3 [25,26,27]. Finally, we divided the time period into daytime and nighttime, based on the 8:00 and 20:00 times, though there may be children who do not sleep during the nighttime, or who take a nap during the daytime. Although more interesting results may have been expected if actual sleep had been reflected, evaluating the actual sleep was difficult due to the retrospective nature of our study.

## 5. Conclusions

We evaluated the clinical significance of changes in HR and RR differentiating between daytime and nighttime and analyzed the association of the changes with CPR occurrence and ICU transfer, demonstrating that there was no clinically relevant association. However, due to the inherent limitations of a retrospective and single-center study, these results may be difficult to apply universally; thus, well-designed further studies are needed to confirm our findings.

## Figures and Tables

**Figure 1 children-10-00167-f001:**
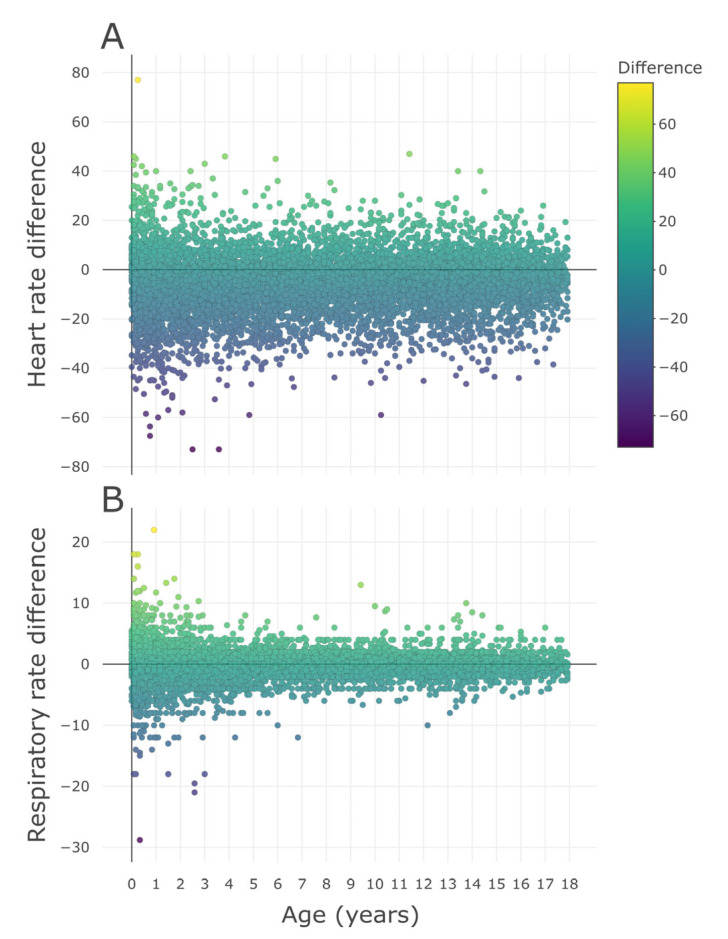
Differences in measurements by age. (**A**) HR difference and (**B**) RR difference by age. The difference of each measurement was defined as the nighttime measurement minus the daytime measurement. Daytime was defined as 8:00 to 20:00, and nighttime was defined as 20:00 to 8:00 the next day. HR = heart rate, RR = respiratory rate.

**Figure 2 children-10-00167-f002:**
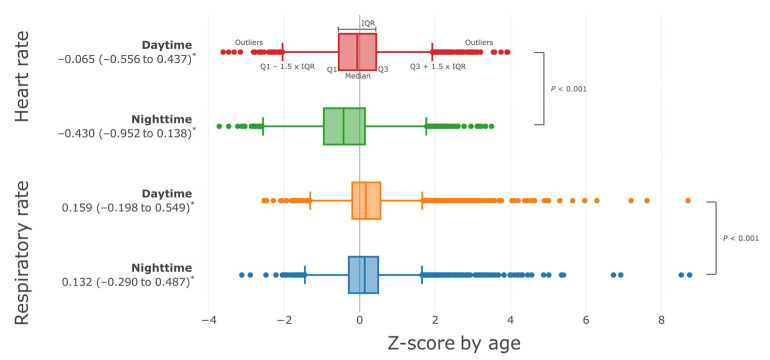
Distribution of z-scores by age for heart rate and respiratory rate by time period. The z-score by age was calculated based on the distribution of heart rate and respiratory rate derived from an existing nationwide study [19]. Daytime was defined as 8:00 to 20:00, and nighttime was defined as 20:00 to 8:00 the next day. Wilcoxon signed-rank test was used for comparison by paired group. * Median (IQR), IQR = interquartile range, Q1 = 1st quartile, Q3 = 3rd quartile.

**Table 1 children-10-00167-t001:** Baseline characteristics of the study population.

Variables	No. of Patients (%) (n = 9778)
Age (years)	7.1 (2.7 to 12.2)
Female sex	4394 (44.9)
Heart rate (beats/min)	
Daytime ^1^	104.0 (92.0 to 119.0)
Nighttime ^2^	98.5 (86.5 to 112.0)
Difference ^3^	−5.1 (−11.7 to 0.5)
Respiratory rate (breaths/min)	
Daytime ^1^	23.5 (20.4 to 27.0)
Nighttime ^2^	23.5 (20.0 to 26.5)
Difference ^3^	0.0 (−1.0 to 0.5)
RRS activation	1021 (10.4)
CPR occurrence	69 (0.7)
ICU transfer	541 (5.5)

Continuous variables are expressed as median (interquartile range). ^1^ Defined as from 8:00 to 20:00; ^2^ Defined as from 20:00 to 8:00 the next day; ^3^ Defined as the nighttime measurement minus the daytime measurement for each individual patient; RRS = rapid response system, CPR = cardiopulmonary resuscitation, ICU = intensive care unit.

**Table 2 children-10-00167-t002:** Analysis of the association of critical event occurrence for each group.

Variables	OR (95% CI)	*p*
Decreased HR group		
RRS activation	4.928 (3.406–7.129)	<0.001
CPR occurrence	0.595 (0–49,800,930.118)	0.956
ICU transfer	0.495 (0.177–1.389)	0.182
Decreased RR group		
RRS activation	2.366 × 10^37^ (NA)	1.000
CPR occurrence	0 (NA)	1.000
ICU transfer	0 (NA)	1.000
Increased HR group		
RRS activation	4.429 (1.155–16.990)	0.030
CPR occurrence	0 (NA)	1.000
ICU transfer	0 (NA)	1.000
Increased RR group		
RRS activation	7.608 (1.788–32.367)	0.006
CPR occurrence	61.58 (0.013–296,863.903)	0.341
ICU transfer	0.078 (0.001–10.379)	0.307

For the analysis, mixed effect model logistic regression analysis was used. The analysis was made with reference to the non-occurrence of each item event. OR = odds ratio, HR = heart rate, RRS = rapid response system, CPR = cardiopulmonary resuscitation, ICU = intensive care unit, RR = respiratory rate, NA = not applicable.

## Data Availability

The data presented in this study are available on request from the corresponding author. The data are not publicly available because it is the policy of the Institutional Review Board of Seoul National University Hospital to destroy the research data after a certain period of time.

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
