# Peer review of "Effect of Diurnal Variation of Heart Rate and Respiratory Rate on Activation of Rapid Response System and Clinical Outcome in Hospitalized Children"

_children, 2023, doi:10.3390/children10010167_

Round 1
Reviewer 1 Report
Dear Authors
Very interesting article, but you should pay attention to some points.
1) there should be no acronyms in the abstract.
2) the first paragraph of the introduction needs more analysis.
3) explain the definitions of the key variables.
4) how the diversity of rhythms between patients of different diseases was evaluated?
5) what is special about this study since its results have already been known for several years?
Author Response
Very interesting article, but you should pay attention to some points.
→ Thank you very much for your review of the paper and affirmative comments of our paper. We supplemented this manuscript more clearly reflecting the reviewer’s comment.
There should be no acronyms in the abstract.
→ Thank you for your helpful comments and we revised the abstract to have no acronyms.
The first paragraph of the introduction needs more analysis.
→ Thank you for bringing out this important point. As your opinion, we explained the first paragraph of the introduction in more detail, especially why heart rate and respiratory rate are important factors.
Explain the definitions of the key variables.
→ Thank you for your helpful comments. We planned this study to evaluate the association of clinical outcomes according to the diurnal variations in vital signs, and analyzed the clinical outcomes with ‘cardiopulmonary resuscitation (CPR) occurrence’ and ‘intensive care unit (ICU) transfer’, respectively. In the manuscript, these two variables are expressed as ‘critical events’, and the definition of them is defined in the part of methods (especially sections of 2.2 and 2.3). However, we added an additional explanation in case it is not enough to understand. Thanks to your advice, we are grateful to be able to define a more accurate definition of our key variables.
How the diversity of rhythms between patients of different diseases was evaluated?
→ As your comment, we conducted an additional statistical analysis of diversity of rhythms between patients of different disease, and the results were summarized as a table in the supplements. Depending on the underlying disease, the difference in vital signs showed significant results in some cases. However, we thought that we should be careful in interpreting this. This is because although we analyzed based on the recorded diagnoses, it was not possible to control the analysis of other variables to see if the difference in the results was due to how the underlying disease actually affected HR or RR.
What is special about this study since its results have already been known for several years?
→ There have been several previous studies describing diurnal variations in vital signs, but there were no results reflecting the effect on the reference ranges or association with clinical outcomes. To the best of our knowledge, this is the first study to analyze each distribution by dividing daytime and nighttime period, and evaluate the clinical significance of changes in vital signs in children. As mentioned in the part of the discussion, it will be meaningful in that this study will broaden medical knowledge and serve as a foundation for further studies. Therefore, the special point of our study was described in paragraph 7 of the discussion part, and we would greatly appreciate it if you could refer to it. Thank you for your helpful comments.

Reviewer 2 Report
This is an interesting and well-written manuscript. I believe the lack of clinical significance is just as important as the opposite. Indeed, the analysis of HR and RR variability during sleep-awake cycle would have probably more significance, but what the authors did is probably the next best thing.
I only have an issue with lines 241-244. They seem written in a novel-like manner that I find inappropriate for this type of paper and clashing with the style of the rest of the manuscript.
Author Response
This is an interesting and well-written manuscript. I believe the lack of clinical significance is just as important as the opposite. Indeed, the analysis of HR and RR variability during sleep-awake cycle would have probably more significance, but what the authors did is probably the next best thing.
→ Thank you very much for your review of the paper and affirmative comments of our paper.
I only have an issue with lines 241-244. They seem written in a novel-like manner that I find inappropriate for this type of paper and clashing with the style of the rest of the manuscript.
→ We appreciate your valuable comments. However, we had some doubts. Because lines 241-244 of the manuscript we finally submitted were “However, even in PEWS, since diurnal variation is not considered in the criteria of HR or RR, although the effect is small, it is thought to have the same inborn limitations.
Some readers may question whether it is necessary to create each criterion for day…”, it didn’t immediately make sense that this read like a novel. And we thought that the sentence you said might be the sentence below.
“Some readers may question whether it is necessary to create each criterion for day and night according to diurnal variation. And some readers may ask what the definition of night and day should be, and what to do if it is nighttime but the child is awake and not sleeping. We deeply sympathize with these concerns.” Even if we read it, we thought that the sentence below was like a novel. At this time, we are not sure if the part you said is the above sentence, but even from our point of view, the above sentence can definitely look like a novel, so we modified the sentence as follows. Thanks for your comment. If we give you an answer that is different from what you intended, please tell us right away. Thank you.
“However, we do not think that it is best to present the reference ranges for daytime and nighttime respectively and apply RRS or PEWS according to the criteria.”
Thank you very much for your time and consideration of our submission. We hope that we have satisfactorily addressed all comments. We hope that our revisions sufficiently improved our manuscript and that it is now suitable for publication.

Round 2
Reviewer 1 Report
Dear Authors
You did a great job! Congratulations!
Author Response
Thank you for reviewing our manuscript, as well as constructive criticism and words of encouragement.